

# Evaluate the Impact of Power-Law Scattering Amplitude Fitting on Dual-Polarization Radar Data Assimilation—Summertime Cases Study

Kao-Shen Chung[1], Chin-Chuan Chang[1], Bing-Xue Zhuang[2], Chih-Chien Tsai[3], Chen-Hau Lan[1,4] and Wei-Yu Chang[1]

[1]Department of Atmospheric Sciences, National Central University, Taoyuan City, Taiwan
[2]Department of Atmospheric and Oceanic Sciences, McGill University, Montreal, Quebec, Canada
[3]National Science and Technology Center for Disaster Reduction, New Taipei City, Taiwan
[4]National Center for Atmospheric Research, Colorado

*Correspondence to: Dr. Kao-Shen Chung (kaoshen.chung@gmail.com)*

**Abstract.** Different configurations within the observation operator cause dual-polarization radar parameters to exhibit various characteristics, which affect the structure of background error covariance as well as the results of data assimilation. Through real case data assimilation experiments, this study evaluates the raindrop-contributed term in the simulated reflectivity ($Z_{HH}$) and differential reflectivity ($Z_{DR}$) to describe the effect of different calculation methods within the operator: the fitting and direct integration methods. In the fitting method, dual-polarization variables are calculated using an analytic function, which assumes a gamma-shaped drop size distribution and fits the relationship between the scattering amplitude (SA) and drop size. In the direct integration method, the quantities of the hydrometeor species and SA are integrated with respect to drop size during the calculation. The results indicate that the fitting method effectively simulates the $Z_{HH}$. However, the limitations of the fitting function may impact the accuracy when represents the structure of $Z_{DR}$. By contrast, the direct integration method effectively simulates polarimetric variables. Validation of the raindrop mass-weighted mean diameter ($D_{mr}$) indicates that assimilation of dual-polarization radar data into the model results in adjustment of the raindrop size distribution regardless of which configuration is used. However, the $D_{mr}$-$Z_{DR}$ structure is closer to the observed structure, and the $Z_{DR}$ structure is more reasonable when the direct integration method is employed. In summary, different configurations within the operator directly affect the results of data assimilation, and the direct integration method has more reasonable performance with respect to simulating dual-polarization radar variables.

## 1 Introduction

In describing the complex changes that occur between hydrometeor species, the cloud microphysics analyses play a crucial role in understanding the growth and decay mechanisms underlying heavy rainfall events. However, because observations are often surface-level and therefore insufficient, accurately representing microphysical structures by using in situ observation data is difficult. Remote sensing provides the alternative solution to this difficulty; in remote sensing, electromagnetic wave propagation is analyzed to observe the bulk features of hydrometeor species. Weather radar with high spatial-temporal resolution offers an advantage in the monitoring of rapidly changing systems and construction of three-dimensional meteorological structures. On traditional Doppler weather radar, precipitation intensity is depicted in terms of radar reflectivity ($Z_{HH}$), and dynamic structures are illustrated in terms of radial velocity ($V_r$). Dual-polarization radar, in which observations are made using two perpendicular electromagnetic waves, provides more information than does traditional radar, leading to a more comprehensive determination of cloud microphysics structures. For instance, differential reflectivity ($Z_{DR}$), which is derived by the reflectivity in different polarimetric directions, describes the shape of hydrometeors inside the rainfall system. Additionally, the specific differential phase ($K_{DP}$), which is estimated from the phase-shift between two polarimetric



directions, closely corresponds to the rain rate and liquid water content. Furthermore, the co-polar correlation ($\rho_{hv}$) indicates

the purity of hydrometeors and detects the non-meteorological signal (Herzegh and Jameson 1992; Zrnic and Ryzhkov 1999), which is useful for radar data quality control (QC). The characteristics of dual-polarimetric variables can be leveraged to construct the three-dimensional structure of hydrometeor species. The interaction between microphysics and dynamics such as the lifting, falling, and size sorting of hydrometeors can also be determined (Kumjian and Ryzhkov 2008; Dawson et al. 2014). For example, a strong updraft may collocate with the $Z_{DR}$ and $K_{DP}$ column because of the lifting of liquid water and the

melting of the mixed-phase particles.

Microphysics parameterizations (MP) have been widely employed for describing complex and nonlinear microphysics processes in the simulation. The two major types of MP schemes— spectral microphysics schemes (SMS) and bulk microphysics schemes (BMS)—are different in terms of how they describe the drop size distribution (DSD) of hydrometeors. SMSs divides drop sizes into different intervals before the DSD is simulated. This approach provides a more flexible

description of the shape of the distribution compared to BMS. However, depicting the DSD in detail may incur high computational costs. In contrast, BMSs employ the gamma-form function (Ulbrich, 1983) to describe the DSD of each type of hydrometeor, leading to lower computational costs and an approximate but acceptable simulation of the DSD. In BMSs, three variables are used to constrain the gamma-form distribution: the intercept parameter ($N_0$), shape parameter ($\mu$), and slope parameter ($\Lambda$). The common DSD formula is as follows:

$$N_X(D_X) = N_{0,X}D_X^\mu \exp(-\Lambda_X D_X) \tag{1}$$

The subscript $X$ represents the hydrometeors, $D$ is the hydrometeor drop size, and $N(D)$ is the number concentration of hydrometeor, which is the function of drop size. The single-moment BMS and double-moment BMS are commonly used in research and operational forecasts. In the single-moment BMS, the model only predicts the slope parameter; the intercept parameter is fixed as a constant or the function of the slope parameter. In comparison with single-moment schemes, double-

moment BMSs can predict slope parameter and intercept parameter, enhancing the number of degrees of freedom in simulating the DSD. The characteristics of simulation with different schemes result from the various configuration of the thresholds, formulas, and constants. For example, to constrain the maximum value of terminal velocity, some schemes fix the minimum of the raindrop slope parameter, which also limits the maximum mean diameter of the raindrop calculated by the moment method. Therefore, the features of different schemes should be validated and considered before any analysis is performed.

Although dual-polarization radar data and model simulation can describe the microphysical evolution of hydrometeors during rainfall periods, the limitations of MP schemes lead to simulation errors, and the radar observations may have uncertainty, even when subjected to quality control. To theoretically reduce the errors and come close to the true state, data assimilation (DA) should be employed to statistically adjust the model toward the true state with the weightings made by observation and background error structure, providing more accurate initial conditions for numerical forecasts. Doppler radar

data, including reflectivity and radial velocity data, have been assimilated using variational method (VAR) or ensemble Kalman filter (EnKF) to capture the rapid evolution of a system (Sun and Crook, 1997; Snyder and Zhang, 2003; Zhang et al., 2004; Xiao et al., 2005; Chung et al., 2009; Chang et al., 2014; Chang et al., 2016; You et al., 2020). Compared to the VAR, the EnKF uses ensemble simulations to generate the background error structure that makes the flow-dependent structure without backward integration. Furthermore, with the EnKF, innovation could be transferred from observation space to model space

without the need to use an adjoint operator, which renders the EnKF suitable for complex and non-linear observation operators, especially dual-polarization radar observation operator. As the dual-polarization radar has matured, the development of dual-polarization radar observation operators (Jung et al., 2008a, 2010; Pfeifer, 2008; Ryzhkov et al., 2011; Augros et al., 2013;



Kawabata et al., 2018; Oue et al., 2020; Shrestha et al., 2022), model validation (You et al., 2020; Shrestha et al., 2022) and data assimilation (Putnam et al., 2019, 2021; Tsai and Chung, 2020) become popular focuses of research. By leveraging the

cross-covariance error structure, the EnKF can adjust several variables in different spaces. For example, Putnam et al. (2019) reported that assimilating only low-level $Z_{DR}$ into the model could further adjust the model at middle and high levels where $Z_{DR}$ is not assimilated. The benefits of assimilating dual-polarization radar data have been evaluated in the literature, with this including adjustment to microphysics structures (Putnam et al. 2019, 2021), improvement of dynamic simulations (Putnam et al. 2019), and, enhancement of rainfall or hail event predictions (Putnam et al., 2019, 2021; Tsai and Chung, 2020).

Electrometric wave scattering caused by hydrometeors plays a key role in radar observation and forward model simulation. The reflectivity factors contributed by hydrometeor scattering can be described as a the function that integrates the numbers of certain particles and the scattering amplitude with respect to drop size (Holt, 1984; Pfeifer, 2008; Jung et al., 2010; Ryzhkov et al., 2011; Augros et al., 2013; Oue et al., 2020). However, the numerical integration may also incur considerable computational cost. To reduce this cost, a function can be used to fit the relationship between the scattering amplitude and drop

size. Subsequently, if the shape of the DSD curve is assumed, the integration formula can be written as an analytic function (Zhang et al., 2001; Jung et al., 2008a; Kawabata et al., 2018). Two types of operators have been employed to assimilate the dual-polarization radar data in the previous studies. However, whether the performance of the fitting operator is consistent with direct integration and the effect of using different operators on data assimilation remain to be evaluated and quantified. In this study, simulated reflectivity and differential reflectivity have been validated to describe the influence caused by different

operator settings. The configurations of the two operators have similar frameworks, but different raindrop contribution terms: one uses an analytic formula after scattering amplitude fitting, whereas the other employs direct integration in calculating. To quantify the effect on data assimilation analyses and forecasts, this study conducts real case DA experiments with different types of summertime rainfall events. The remainder of this paper is organized as follows. Section 2 introduces the data assimilation system and the forward operators that are implemented in this study. Section 3 describes the environment of the

case and the observation data that are used. The configurations of model and experiments are introduced in section 4. Results and discussion are presented in section 5; and the summary and future work are detailed in the section 6.

## 2 Data Assimilation System and Observation Operators

### 2.1 WRF-LETKF Radar Assimilation System (WLRAS)

In this study, the ensemble-based data assimilation system, WRF-LETKF Radar Assimilation System (WLRAS, Tsai et

al., 2014) is employed to assimilate dual-polarization radar observation data. Based on Local Ensemble Transform Kalman Filter (LETKF; Ott et al., 2004; Hunt et al., 2007), WLRAS is a deterministic EnKF that does not inquire about the perturbation of observations; this approach avoids the sampling errors during assimilation. In practice, the WLRAS would update the ensemble mean and perturbation separately by using the weighting constructed from the observation-space background error covariance and observation variance. The formulas are as follows:

$$\overline{\mathbf{X}_a} = \overline{\mathbf{X}_b} + \mathbf{X}_b' \widetilde{\mathbf{P}_a} \mathbf{Y}_b'^T \mathbf{R}^{-1} (\mathbf{y}_o - \mathbf{y}_b) \qquad (2)$$

$$\mathbf{X}_a' = \mathbf{X}_b' [(K-1)\widetilde{\mathbf{P}_a}]^{\frac{1}{2}} \qquad (3)$$

$$\widetilde{\mathbf{P}_a} = [(K-1)\mathbf{I}/\rho + \mathbf{Y}_b'^T \mathbf{R}^{-1} \mathbf{Y}_b']^{-1} \qquad (4)$$

The subscripts $a$ and $b$ denote analysis and background status, respectively. $\overline{X}$ and $\mathbf{y}_b$ are the ensemble mean vector in the model space and observation space, respectively. $X'$ is the ensemble perturbation matrix, $\mathbf{y}_o$ is the observation state vector.



$Y'_b$ is the background ensemble perturbation matrix, which is interpolated into observation space. $R$ is the observation error covariance matrix, which is usually considered an error variance matrix when observations are assumed to be independent. The background error covariance may be converted into the analyses error covariance by using the transformation matrix $\widetilde{P_a}$, in which $K$ is the ensemble size and $I$ is the identity matrix. The multiplicative inflation factor ($\rho = 1.08$) is included during assimilation and ensures that the ensemble has sufficient spread to avoid the filter divergence. The mixed localization

radius method (Tsai et al., 2014) being applied in WLRAS ensures a reasonable updated range of different state variables with their own scale characteristic. Furthermore, considering the dynamic imbalance caused by extreme adjustment during the DA cycling, the decision regarding whether an observation is assimilated is made using a threshold of innovation at each observed position; the radar data would be assimilated when the innovation is less than three times the observation standard deviation.

   In this study, the employed configuration of the WLRAS system, including the localization radius and multiple inflation

factors, is set on the basis of that used in You et al. (2020) and is presented in Table 1. The observation error standard deviation of the radial velocity, reflectivity, and differential reflectivity are considered to be 2 m s$^{-1}$, 5 dBZ (You et al. 2020), and 0.2 dB (Jung et al. 2008b), respectively. Variable localization is conducted when assimilating different observed parameters into the model. Finally, the super-observation method is performed on each elevation to prevent overfitting during assimilation. The resolution in the super-observation method is 2 km and 2°, and this method is applied at each observed elevation.


### 2.2 Observation Operator

   Since dual-polarization parameters cannot be prognosticated by a numerical model, variables must be transformed before calculating the error covariance and the innovation. To directly assimilate radar data without variable retrieval, the observation operator can transform the prognostic variables into simulated radar variables on the basis of observation theories. In this study,

the observation operators of the radial velocity and dual-polarization parameters are implemented; these operators are described in the following subsections.

#### 2.2.1 Radial velocity observation operator

   The observation operator of the radial velocity considers three-dimensional wind and hydrometeor terminal velocity. The

following formula is used for calculating the radial velocity:

$$V_r = \frac{Ux + Vy + (W - V_t)z}{(x^2 + y^2 + z^2)^{0.5}} \tag{5}$$

where $U$, $V$, and $W$ are the three-dimensional wind components, and $x$, $y$, and $z$ represent the three-dimensional components of the distance between particle and radar in Cartesian coordinates. The raindrop terminal velocity ($V_t$) is also included in calculating the radial velocity; the relevant formula is as follows:

$$V_t = 5.4(\rho_a q_r)^{0.125} \left(\frac{p_0}{\bar{p}}\right)^{0.4} \tag{6}$$

where $\rho_a$ is the density of air; $q_r$ is the rainwater mixing ratio, and $p_0$ is the surface pressure. Furthermore, $\bar{p}$ is the average pressure in a certain layer and is used to calculate $V_t$ at different altitudes.

#### 2.2.2 Dual-polarization observation operator

   In this study, the dual-polarization observation operator based on polarimetric radar data simulator (PRDS; Jung et al.,



2008a, 2010) is implemented. The following equations reveal the configuration of the operator. First, the horizontal and vertical reflectivity factors ($Z_{h,x}$ and $Z_{v,x}$) of each hydrometeor species are derived by Eq. (7) and (8):

$$Z_{h,x} = \frac{4\lambda^4}{\pi^4|K_w|^2} \int N_x(D)(A|f_{h,x}(\pi)|^2 + B|f_{v,x}(\pi)|^2 + 2C|f_{h,x}(\pi)||f_{v,x}(\pi)|)dD \tag{7}$$

$$Z_{v,x} = \frac{4\lambda^4}{\pi^4|K_w|^2} \int N_x(D)(A|f_{v,x}(\pi)|^2 + B|f_{h,x}(\pi)|^2 + 2C|f_{h,x}(\pi)||f_{v,x}(\pi)|)dD \tag{8}$$

where $N_x(D)$ is hydrometeor number concentration, and $K_w$ is the dielectric factor of water; $|f_{h,x}(\pi)|$ and $|f_{v,x}(\pi)|$ are

horizontal and vertical backscatter amplitude, respectively. The hydrometeor canting effect is described by the coefficients $A$, $B$ and $C$, which are calculated using the following equations:

$$A = \frac{1}{8}(3 + 4\cos 2\bar{\Phi} e^{-2\sigma^2} + \cos 4\bar{\Phi} e^{-8\sigma^2}) \tag{9}$$

$$B = \frac{1}{8}(3 - 4\cos 2\bar{\Phi} e^{-2\sigma^2} + \cos 4\bar{\Phi} e^{-8\sigma^2}) \tag{10}$$

$$C = \frac{1}{8}(1 - 4\cos 4\bar{\Phi} e^{-8\sigma^2}) \tag{11}$$

where $\bar{\Phi}$ is the mean and $\sigma$ is the standard deviation of the canting angle. Considering the scattering of mixed-phase particles, the PRDS establishes the mixing model with the fraction of ice particles and rainwater. Also, the characteristics of mixing particles, such as the canting motion and the scattering intensity, are described in the mixing model. The coefficients and formula of the mixing-phase model are established as they are in You et al. (2020) and Jung et al. (2008a). Finally, the reflectivity and differential reflectivity in decibel scale can be obtained from the combination of reflectivity factors of all

hydrometeor species as follows:

$$Z_{HH} = \log_{10}(\sum Z_{h,x}) \tag{12}$$

$$Z_{VV} = \log_{10}(\sum Z_{v,x}) \tag{13}$$

$$Z_{DR} = \log_{10}(\sum Z_{h,x} / \sum Z_{v,x}) \tag{14}$$

### 2.2.3 Scattering amplitude inside the observation operator

In Eqs. (7) and (8), the reflectivity factor can be derived by integrating total number concentrations and the backscattering amplitude with respect to diameter. To calculate the scattering amplitude, previous studies utilized T-matrix scattering calculations (Waterman, 1969; Vivekanandan et al., 1991) before simulating the dual-polarization parameters. The T-matrix method considers the impact of the canting motion and the environment on electrometric wave transmission to ensure the simulation is as real as possible. The setting of environmental parameters, including the background temperature and density

of hydrometeor species, may lead to different refraction characteristics. In addition, the particle axis ratio is fixed for each hydrometeor species to describe the characteristic scattering caused by the shape of the hydrometeors. A look-up table, which records the scattering amplitude for different diameter intervals, can be created from the results of single-particle simulation by using the T-matrix method. The reflectivity factors can then be derived using Eq. (7) and (8). Many of the operators have a similar framework (Pfeifer, 2008; Jung et al., 2010; Ryzhkov et al., 2011; Augros et al., 2013; Oue et al., 2020). Although the

integration method may yield dual-polarimetric parameters with reasonable structure, its computational cost is inevitably high. To decrease this cost, analytic equations are used in some operators to replace the integration (Zhang et al., 2001; Jung et al., 2008a; Kawabata et al., 2018). In these operators, the scattering amplitude is described as a function of drop size with





assumptions, such as the function form of the hydrometeor axis ratio. Additionally, using the bulk microphysics scheme, the DSD can be described by gamma function parameters [the intercept, shape, and slope parameter in Eq. (1)]. This approach simplifies the calculation because integration does not need to be performed over the entire diameter interval.


In this study, the main structure of the operator is based on Jung et al. (2008a), which uses an analytic function to obtain the dual-polarimetric parameters. To fit the relationship between the particle diameter and scattering amplitude, the power law function is applied as follows:

$$|f_h| = \alpha_{h,x} D_x^{\beta_{h,x}} \tag{15}$$


$$|f_v| = \alpha_{v,x} D_x^{\beta_{v,x}} \tag{16}$$

where $D$ is the diameter of a hydrometer species and $\alpha_{h,x}$, $\alpha_{v,x}$, $\beta_{h,x}$, and $\beta_{v,x}$, are fitting coefficients that are set to those used in You et al. (2020) and Jung et al. (2008a). Substituting the fitting coefficients and assumed the gamma-shaped DSD into Eqs. (6) and (7) yields the following equations:

$$Z_{h,x} = \frac{4\lambda^4 N_{0,x}}{\pi^4 |K_w|^2} (A\alpha_{h,x}^2 \frac{\Gamma(\mu_x + 2\beta_{h,x} + 1)}{\Lambda_x^{\mu_x + 2\beta_{h,x} + 1}} + B\alpha_{v,x}^2 \frac{\Gamma(\mu_x + 2\beta_{v,x} + 1)}{\Lambda_x^{\mu_x + 2\beta_{v,x} + 1}} + 2C\alpha_{h,x}\alpha_{v,x} \frac{\Gamma(\mu_x + \beta_{h,x} + \beta_{v,x} + 1)}{\Lambda_x^{\mu_x + \beta_{h,x} + \beta_{v,x} + 1}}) \tag{17}$$


$$Z_{v,x} = \frac{4\lambda^4 N_{0,x}}{\pi^4 |K_w|^2} (A\alpha_{v,x}^2 \frac{\Gamma(\mu_x + 2\beta_{v,x} + 1)}{\Lambda_x^{\mu_x + 2\beta_{v,x} + 1}} + B\alpha_{h,x}^2 \frac{\Gamma(\mu_x + 2\beta_{h,x} + 1)}{\Lambda_x^{\mu_x + 2\beta_{h,x} + 1}} + 2C\alpha_{h,x}\alpha_{v,x} \frac{\Gamma(\mu_x + \beta_{h,x} + \beta_{v,x} + 1)}{\Lambda_x^{\mu_x + \beta_{h,x} + \beta_{v,x} + 1}}) \tag{18}$$

The shape parameters ($\mu$) of different hydrometeor species are fixed using the same configuration as that in the WRF double-moment 6-class (WDM6) MP scheme (Lim and Hong, 2010). The slope parameter ($\Lambda$) and intercept parameter ($N_0$) can be estimated using the following equations:

$$\Lambda_x = [\frac{\pi \rho_x N_{T,x} \Gamma(\mu_x + 4)}{6\rho_a q_x \Gamma(\mu_x + 1)}]^{\frac{1}{3}} \tag{19}$$


$$N_{0,x} = \frac{\Lambda_x^{\mu_x + 1} N_{T,x}}{\Gamma(\mu_x + 1)} \tag{20}$$

where $\rho_a$ is the density of air; $\rho_x$ is the density of hydrometeor species in the corresponding setting of the MP scheme. $q_x$ and $N_{T,x}$ are the predicted mixing ratio and total number concentration of hydrometeor species, respectively. Finally, by using Eqs. (17) and (18), the reflectivity factor can be derived without integration. The details of the configuration in the PRDS are described in Jung et al. (2008a).


The direct integration method can also be employed in calculating the raindrop-contributed reflectivity factor and is compared with the fitting method in this study. The T-matrix method is applied to simulate the scattering amplitude with respect to different raindrop sizes. The environmental conditions of the T-matrix simulation, including temperature, canting angle, and drop size bin intervals, are set on the basis of the setting in Jung et al. (2010) and are preasented in Table 2. Finally, the factors of the raindrop-contributed reflectivity factors on both axes are derived by directly integrating the raindrop number

concentration and scattering amplitude with respect to the drop size diameter.




## 3 Data and Case Overview

### 3.1 Case Overview

In this study, real case data assimilation experiments on summertime rainfall events are conducted to evaluate the influence of the employed specific operator configuration. Three cases are selected, including a squall line case, a typhoon case and a Mei-Yu frontal case (stationary frontal case), to validate the simulations for various weather types.

### 3.1.1 Squall line case

Located near the boundary of Eurasia and the Pacific Ocean, Taiwan has the weather system that is strongly affected by the moisture in air transported by the East Asia monsoon, especially in terms of its summertime rainfall. This study selects the case of squall line on June 14, 2008, during the eighth intensive observation period of the Southwest Monsoon Experiment (SoWMEX-IOP8). In this period, the southeastern Asian monsoon low drove a southerly wind near Taiwan and propagated moist air from the Indian Ocean to the South China Sea and Taiwan Strait, creating an environment favorable for system development. Therefore, several mesoscale systems, including the squall line system, developed and impacted southern Taiwan, causing more than 300 mm of rainfall over three days (Lupo et al., 2020). At 1200UTC on June 14, 2008, the squall line system was well developed and impacted southern Taiwan. As revealed by the National Center of Atmospheric Research (NCAR) S-band dual-polarimetric radar (S-POL) observation, $Z_{HH}$ extended to higher than 45 dBZ, and $Z_{DR}$ was mainly within 0–1 dB. Thus, the strong rainfall system mainly comprised small raindrops.

### 3.1.2 Typhoon case

During summer and autumn, Taiwan is frequently struck by typhoons. On average, approximately three typhoons affect the island each year. They bring destructive winds and considerable rainfall, often destroying the landscape and leading to severe flooding. Therefore, studies of typhoons is crucial for Taiwan. In this study, a violent typhoon, Typhoon Soudelor, is selected as the research target to validate the impact of dual-polarization data assimilation. Typhoon Soudelor affected Taiwan from August 7 to 9, 2015, and caused more than 300 mm of rainfall during this period. Some places, particularly mountainous regions, received more than 600 mm of rainfall. Considerable damage was caused by the typhoon, including flooding and landslides in northern Taiwan. Furthermore, a record-breaking gust of wind devastated the electric power system, leading to the largest blackout on record in Taiwan. The radar observation made by the Wufenshan Weather Radar Station (RCWF) indicated that an intense rain band of more than 45 dBZ directly struck northeastern Taiwan and resulted in heavy rainfall. The high occurrence frequency of $Z_{HH}$ extended to 45 dBZ, and $Z_{DR}$ was mainly within 0.5–1.3 dB, indicating that the rain band mainly comprised small raindrops but larger drops than those in the squall line case and Mei-Yu frontal case.

### 3.1.3 Mei-Yu (stationary frontal MCS) case

The East Asia rainy season, also referred to as the Mei-Yu season, typically occurs during the early summer in middle-eastern Asia and late summer in northeastern Asia. The quasi-stationary front caused by cold and dry air from the north and warm and moist air from the south leads to an unstable environment, inducing an intense mesoscale convective system and sustained rainfall. The frontal system on June 6, 2022, during the third intense observation period of the Taiwan-Area Heavy Rain Observation and Prediction Experiment (TAHOPE) is selected in this study. The frontal system coupled with the low-level moisture environment gradually approached Taiwan, and a linear convective precipitation system was located in the north of Taiwan at 0800 UTC. Radar observation by NCAR SPOL and RCWF indicated that the high occurrence frequency of $Z_{HH}$ extended to 40 dBZ, and that $Z_{DR}$ was mainly within 0.5–1.0 dB, indicating that the rain band was mainly comprised of



raindrops.

### 3.2 Radar Data Quality Control

In this study, the radar data from NCAR SPOL and operational S-band weather radars— RCWF (Wufengshan Radar Weather Station), RCKT (Kenting Radar Weather Station), RCHL (Hualien Radar Weather Station), and RCCG (Chigu Radar Weather Station)—are assimilated into the model. Since the radar observation network has gradually updated from single-polarization to dual-polarization radar, the observed parameters and the scanning strategies of radars are different in three cases. In the SoWMEX IOP8 squall line case, all operational radars had the same scanning strategies: nine elevation angles (0.5°,

1.4°, 2.4°, 3.4°, 4.3°, 6.0°, 9.9°, 14.6° and 19.5°) and provided single-pol Doppler radar parameters, the reflectivity ($Z_{HH}$) and radial wind ($Vr$). For the cases after 2015, the same strategies were used for most radars, although RCWF was updated to dual-polarization weather radar, providing operational dual-polarization radar data with 15 elevation angles in the one volume scan (0.5°, 0.9°, 1.3°, 1.8°, 2.4°, 3.1°, 4.0°, 5.1°, 6.4°, 8.0°, 10.0°, 12.0°,14.0°, 16.7°, and 19.5°). NCAR SPOL data were used in the 2008 SoWMEX and 2022 TAHOPE, and the radar provides dual-polarization data. SPOL scanned with 9 elevation angles

(0.5°, 1.1°, 1.8°, 2.6°, 3.6°, 4.7°, 6.5°, 9.1° and 12.8°) in 2008 and 10 elevation angles (0.5°, 1.0°, 1.5°, 2.0°, 3.0°, 4.0°, 5.0°, 7.0°, 9.0° and 11.0°) in 2022, providing substantial data regarding the nearby surface, and therefore, the structure of the intense rainfall system could be comprehensively studied.

A quality control process is used to prevent the model from being contaminated by non-meteorological signals. On the basis of the characteristics of dual-polarization radar parameters, certain thresholds are employed to remove noise. The Radar

Kit (Rakit) developed by National Central University is employed to control the quality of the radar data. Single-polarization radar data are first corrected by terrain height to avoid the radar beam being blocked; then, the radial velocity is unfolded and used to remove the non-meteorological signal in accordance with a certain threshold: $Z_{HH} > 30$ dBZ and $Vr < 2$ m s$^{-1}$. For the dual-polarization radars, in the first step of quality control, $\Phi_{dp}$ at different elevations is unfolded. Subsequently, $\rho_{hv} > 0.9$ and $\Phi_{dp}$ standard deviation $< 10$ are applied to filter out non-meteorological noise. In addition, the radial velocity would be unfolded.

Finally, the $Z_{DR}$ systematic bias of RCWF is calculated by the mean value of $Z_{DR}$ observation at the low-reflectivity (15~25 dBZ) region using the method mentioned in Loh. et. al. (2022).

### 4   Configuration of Model and Experiments

### 4.1 Model Configuration

WRF version 3.9.1 is applied to simulate the summertime rainfall events and to conduct radar data assimilation in the

study. Three nested domains are employed. For each domain, the top of the model is set at 10-hPa height with 52 η levels, and the horizontal resolution is 15, 3, and 1km in the first, second, and third domain (D01–D03), respectively. To obtain a reasonable synoptic-scale structure, NCEP FNL operational model global tropospheric analyses with 0.25° resolution are used to generate the initial and boundary conditions. Four physics parameterization schemes are used, including Dudhia short-wave radiation parameterization scheme (Dudhia 1989), Rapid Radiative Transfer Model (RRTM) longwave radiation

parameterization scheme (Mlawer et al. 1997), Yonsei University (YSU) planetary boundary layer parameterization scheme (Hong et al. 2006), and Grell-Freitas cumulus parameterization scheme (Grell and Dévényi 2002; only in D01) to describe the sub-grid physical processes. Moreover, the WDM6 MP scheme (Lim and Hong 2010) is implemented to evolve the performance of the microphysics processes. To conduct EnKF data assimilation, ensemble forecasts with 50 members are generated. For each member, the horizontal wind field, perturbation potential temperature, and water vapor in D01 are

perturbed using the WRFDA cv3 option; then, the perturbed initial condition would soon be downscaled to the nested domains




after perturbation. For all of the cases, the simulation would be initialized and spun up before the first DA cycle.

### 4.2 Experiment Design

Three experiments are conducted in this study (Table 3). In the VrZ experiment, only $Z_{HH}$ and $V_r$ are assimilated, which is used to validate the background performance when $Z_{DR}$ is not assimilated into model. In the VrZZ_FIT experiment, $Z_{HH}$, $V_r$ and $Z_{DR}$ are assimilated using the fitting method inside the operator. In the VrZZ_DIR experiment, the same variables are assimilated but the direct integration method is used to calculate the raindrop-contributed reflectivity factor. The DA procedure in the three cases is illustrated in Fig. 6, including the model spin-up and 2-hour DA window. The period of each DA cycle is 15 minutes in the squall line and the typhoon case, but 12 minutes in the Mei-Yu frontal case because more rapid scanning is available in that event. In each cycle, the dual-polarization radar data are assimilated in the following order: (1) Vr, (2) $Z_{HH}$, and (3) $Z_{DR}$. Finally, only $Z_{DR}$ observations below 4 km are assimilated, which is below the melting layer in the model and observation, and this is performed to specifically focus on the impact of using different calculation methods on the raindrop-contributed term.

### 5    Results and Discussions

### 5.1 Performance of Simulated Variables

The background error covariance between the assimilated variables and state variables is strongly affected by the configurations inside the forward operators. Also, the innovation, which indicates the direction of the model updating, may be incorrect when the simulation has some bias caused by the model and forward operator. Therefore, the performance of the forward operator must be evaluated before data assimilation. In this study, the simulated dual-polarization radar parameters in the background are validated by observation data. Fig. 7 presents the overall performance of the analyzed dual-polarization radar parameters in experiment VrZ below 4 km when two operators are used to transform the variables. The results reveal that both operators capture the structure of reflectivity, in which the mean difference between simulation and observation is less than 5 dBZ for the whole period. The results obtained using different calculation methods are in strong agreement, indicating that the fitting method successfully simulates reflectivity while consuming fewer computational resources. However, the simulated $Z_{DR}$ using the fitting method operator differs considerably from observation, and the uncertainty is higher than that in the VrZ_DIR experiment and the observation. Additionally, an unreasonable negative $Z_{DR}$ is simulated through the fitting method in the Mei-Yu frontal case, and considerable overestimation occurs in the typhoon case. By contrast, directly deriving the dual-polarization parameters through numerical integration not only captures the $Z_{HH}$ structure but also yields a reasonable $Z_{DR}$ in the VrZ experiment, leading to much more $Z_{DR}$ data being assimilated under the same innovation threshold of the DA (Fig. 8).

The reason for the $Z_{DR}$ negative bias can be determined from Eqs. (16) and (17). If we only focus on the raindrop's contribution to the reflectivity, the Eqs. (16) and (17) could be written as follows:

$$Z_{DR,rain} = 10\log_{10}(\frac{Z_{h,rain}}{Z_{v,rain}}) = 10\log_{10}\left(\frac{\Gamma(2\beta_{r,h}+\mu+1)}{\Gamma(2\beta_{r,v}+\mu+1)}\Lambda^{[(2\beta_{r,h}+\mu+1)-(2\beta_{r,v}+\mu+1)]}\right) \tag{21}$$

Subsequently, when the fitting coefficients mentioned by Jung et al. (2008a) are used, $\beta_{r,h}$ and $\beta_{r,v}$ are 3.04 and 2.77, respectively. Substituting these coefficients and the shape parameter of raindrops in WDM6 ($\mu = 1$) into Eq. (23) yields the following:





$$Z_{DR,rain} \approx 10\log_{10}(2.928\Lambda^{-0.54} \times 10^{1.62}) \approx 10\log_{10}(122.08\Lambda^{-0.54}) \tag{22}$$

Due to the law of the logarithm, when $\Lambda^{-0.54}$>122.08, $Z_{DR,rain}$ may be less than 0, leading to the unreasonable negative $Z_{DR,rain}$ value in the simulation. The same result may be obtained for other operators when the power-law function is used to describe the relationship between scattering amplitude and drop size. Hence, before applying the forward operator to simulate
the dual-polarization radar parameters, the properties of the operator should be considered to ensure the simulation is reasonable in the given interval of the drop size.

**5.2 Validation of Dual-Polarization Parameters**

This section details the validation of the simulated variables for EnKF forecasts and analyses. To estimate the difference between observation and simulation data, root mean square residual (RMSR) is calculated as follows:

$$RMSR = \sqrt{\frac{\sum(y^o - H\bar{x}^a)^2}{n}} \tag{23}$$

The term inside the parentheses is the residual, which means the difference between observation and analyses in observation space. The closer the RMSE is to zero, the closer the simulation is to the observation. Fig. 9 shows the $Z_{HH}$ and $Z_{DR}$ RMSR for the entire DA period. To estimate the impact of the raindrop-contributed term, the $Z_{DR}$ RMSR is calculated with the observations below 4 km, which are the height lower than the melting layer in the observation and simulation. The $Z_{HH}$ RMSR values
determined for the two operator settings are similar in the three cases, and the difference between the VrZZ_DIR and VrZZ_FIT experiments is approximately 1.0–2.0 dBZ in all cases. However, the $Z_{DR}$ RMSR indicates a considerable difference between two experiments that using the direct integration method (VrZZ_DIR) has a lower residual compared to VrZZ_FIT. This result can be explained by two reasons: First, the direct integration method prevents negative $Z_{DR}$ bias and yields a more similar $Z_{DR}$ structure during DA cycling. Conversely, much more $Z_{DR}$ observation information can be assimilated into the model and used
to modify the simulation toward the observation when using the direct integration method. The difference joint frequencies calculated from analyses and observation below a height of 4 km (Fig. 10) indicates the similar performance that in the VrZZ_FIT experiment, $Z_{DR}$ is obviously overestimated for the three cases. Some high-frequency difference is even greater than +0.6 dB or less than -0.6 dB, which exceeds the threshold of the assimilation that $Z_{DR}$ cannot be assimilated. Corresponding results could be indicated by the $Z_{HH}$–$Z_{DR}$ joint frequencies under a height of 4 km (Fig. 11). The occurrence frequency in the
background shows that the slope of the high-frequency area in the VrZZ_FIT experiment is greater than the slope in the observation data. By contrast, the $Z_{HH}$–$Z_{DR}$ joint frequency in the VrZZ_DIR experiment is much closer to the observation. Assimilating $Z_{DR}$ into the model results in the frequency being adjusted toward the observed value in both experiments, but the adjustment in the VrZZ_DIR experiment is greater than the one in the VrZZ_FIT experiment.

Although the validations above suggest that the VrZZ_DIR experiment has a more reasonable simulation compared to
the VrZZ_FIT experiment, there is still a difference between the VrZZ_DIR simulation and the observation. For instance, the occurrence frequency in the VrZZ_DIR experiment of $Z_{DR}$ is overestimated for the typhoon case when $Z_{HH}$>30 dBZ. Besides, the joint frequency distribution in the VrZZ_DIR experiment is narrower than the one from observation data, indicating that the operator could not appropriately describe the degree of freedom in the observation. However, the $Z_{DR}$ overestimation and narrow distribution in the VrZZ_DIR experiment could be modified after the assimilation, leading to a more reasonable
structure compared with that obtained in the VrZZ_FIT. In sum, the clear difference in the $Z_{HH}$–$Z_{DR}$ structure in the background limits the adjustment of assimilation in the VrZZ_FIT experiment, and the similar $Z_{HH}$–$Z_{DR}$ structure in the VrZZ_DIR background leads to a more reasonable structure in the analyses.



### 5.3 Performance of Microphysics Structure

Since dual-polarization radar parameters are not prognostic variables, the adjustments of updated variables must be
validated to determine how data assimilation affects the model. To evaluate the microphysics structure, this study uses the
raindrop mass-weighted mean diameter ($D_{mr}$) retrieved from the dual-polarization radar observation as a reference observation.
$D_{mr}$ is obtained through the method reported by Cao et al. (2008) by using 15 years of disdrometer data observed at National
Central University referenced by Lee et al. (2019). The result of fitting is as follows:

$$D_{mr} = Z_{HH}(0.00946Z_{DR}{}^3 - 0.0277Z_{DR}{}^2 + 0.0323Z_{DR} + 0.0338) \tag{24}$$

Fig. 12 and Fig. 13 present the $D_{mr}$–$Z_{DR}$ occurrence joint frequency below 4 km height in the final cycle background and
analyses, respectively. In all experiments, $D_{mr}$ is underestimated in the background but is adjusted toward the observed value
after dual-polarization radar assimilation. According to the comparison of the VrZZ_FIT and VrZZ_DIR experiments, the
difference in $D_{mr}$ between the two experiments is small, but the $D_{mr}$–$Z_{DR}$ structure is more concentrated in the VrZZ_DIR
findings because of the more correct $Z_{DR}$ structure. Thus, when the method employed to calculate the raindrop-contributed
term is changed, the characteristics of the raindrop prognostic variables do not change completely; however, the relationship
between $D_{mr}$ and $Z_{DR}$ is more reasonable when the direct integration method is employed.

### 6    Conclusions

Cloud microphysics describes the interactions between hydrometeor species, which directly influence the growth and
decay of rainfall systems and the intensity of precipitation. In monitoring the rapid development of rainfall systems, dual-
polarization radar, in which electrometric waves in two directions are detected, can be used to obtain comprehensive insight
into cloud microphysics and depict the bulk characteristic of the hydrometeor species. The characteristics of dual-polarization
radar parameters have been widely used in particle identification, radar data QC and depicting the microphysical structure.
Because the dual-polarization radar data are not prognostic variables in the numerical model, variable transformation is
required before model validation and data assimilation can be performed. The observation operator can derive the observed
variable in the observation space by using the predicted variable in the model space, leading to a more direct connection
between the model and observation. Dual-polarization radar operators have been developed and utilized for data assimilation
in several studies. Such operators have two major types of configuration: those using the fitting method and those using the
direct integration method. Both methods calculate dual-polarization parameters by integrating scattering amplitudes over the
given DSD. In the fitting method, the scattering amplitudes are fitted as a function of diameter, enabling an analytical solution
to the integration and improving computational efficiency; while in the direct integration method, the numerical integration is
used to calculate the dual-polarization radar parameters directly. Different simulations of dual-polarization radar parameters
may directly change the structure of the background error covariance, creating various results after DA. Hence, to obtain the
most reasonable and unbiased background structure, simulations of dual-polarization parameters involving different types of
operators should be evaluated. In this study, a dual-polarization radar operator established on the basis of that reported by Jung
et al. (2008a) is used to simulate and assimilate dual-polarization radar data into the model. Additionally, the raindrop-
contributed terms are obtained using the fitting method (Jung et al., 2008a) and the direct integration method (Jung et al. 2010)
to describe the effect of operator configuration on data assimilation. Finally, three types of summertime rainfall events,
including the squall line case, the typhoon case, and the Mei-Yu frontal case (quasi-stationary front), are selected to
comprehensively validate the simulation in the real case.

The evaluation reveals that both configurations can simulate a reflectivity structure similar to that in observations.





However, the simulated $Z_{DR}$ using the fitting method has an unreasonable negative bias below the melting layer, which occurs because of the limitations of power-law fitting. Compared with that obtained using the fitting method, the $Z_{DR}$ simulated using the direct integration method is similar to the observation data, leading to a structure more similar to that in the observation. The same results are obtained from the $Z_{HH}$–$Z_{DR}$ joint frequency diagram. The fitting method leads to negative bias and

overestimation of $Z_{DR}$ when the reflectivity is greater than 30 dBZ. Instead, when the direct integration method is used, the high-occurrence region is closer to the observations than with the fitting method. In this case, the $Z_{DR}$ difference between simulations and observations is limited to within ±0.6 dB. However, the overestimation of $Z_{DR}$ still persists when the reflectivity exceeds 30 dBZ, even with the direct integration method. In addition, it could not comprehensively describe the spread of the joint frequency distribution, which indicates the operator has an insufficient degree of freedom. The performance of raindrop

prognostic variables is validated to describe how dual-polarization radar data assimilation affects the model. The results reveal that assimilating dual-polarization radar parameters into the model do not clearly improve the $D_{mr}$ simulation. However, using the direct integration method rather than the fitting method leads to a more favorable result; the relationship between the $D_{mr}$ and $Z_{DR}$ occurrence frequency is more concentrated and closer to the observed relationship. Crucially, both the forward operator and the MP scheme cannot exactly simulate the details of complex mechanism underlying the mesoscale system, and it may

directly impact the short forecasts during DA cycling, diluting the advantages of the DA. These problems should be further studied to enable modification of the system and establish a suitable DA strategy to enhance the predictability.

To sum up, using the power law function to fit the scattering amplitude may lead to unreasonable negative bias in $Z_{DR}$ simulation. By contrast, using the direct integration method to calculate the raindrop-contributed reflectivity factor can result in a more reasonable $Z_{HH}$ and $Z_{DR}$ structure and an acceptable $D_{mr}$ structure after DA. However, this study also reveals some

problems that have to be further investigated. Because the characteristics of hydrometeor species may differ for various configurations inside schemes, the microphysics processes in the short forecasts must be confirmed, and the most suitable scheme for simulating different weather systems should be identified. In terms of the dual-polarization radar observation operator, simulation related to ice particles has considerable uncertainty because ice particles can have various shapes and densities, and exhibit various canting motions. Finally, since the $Z_{HH}$ and $Z_{DR}$ bias in the background during DA cycling directly

limits the effect of DA, except for the forward operator, the microphysics processes in MP schemes should be validated and optimized using the real time observation to ensure that simulations could preserve the benefits of assimilation during short forecasting.

Next, dual-polarization parameter simulations with different microphysics schemes, particularly the new version of the multi-moment microphysics schemes, should be validated with different weather cases. Additionally, we attempt to validate

the benefits of assimilating different-waveband, non-operational regional weather radar data, which are often used in bean-blockage regions. Regarding the forward operator, it is necessary to test the initial setting of the ice particle T-matrix simulations must be further tested to adjust the ice contribution term to the observation.

*Data availability.* The operational radar data were provided by Central Weather Adiminstration, Taiwan. The meteorological observation data are also available from Taiwan CWA at https://data.gov.tw/en/datasets/9176. The SPOL radar data are

provided by National Center for Atmospheric Research. The NCEP 0.25° reanalysis can be downloaded at https://rda.ucar.edu/datasets/ds083.3/.

*Code availability:* The source code of WLRAS and the forward operator are not available on any public repository.



*Author contributions.* K.-S. C. proposed the idea and conducted the experiments for the tests, then supervised丶constructed and initialized the draft of the manuscript. C.-C. C. completed the draft of the manuscript and performed the formal analysis,
software coding, and visualization of the results. B.-X. Z, C.-C. T., L.-C. H. and W.-Y. C. provided T-matrix source code, interpreted and discussed the data results. All authors contributed to the final paper.

*Competing interests.* The authors declare no conflicts of interest.

*Acknowledgements:* This study was funded by the National Science and Technology Council of Taiwan through Research Grant 111-2111-M-008-023. The authors wish to express their gratitude to the National Center of High-Speed Computing
and Atmospheric Science Research and Application Databank for providing the authors with the computational resources and observation data from the Central Weather Administration.

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



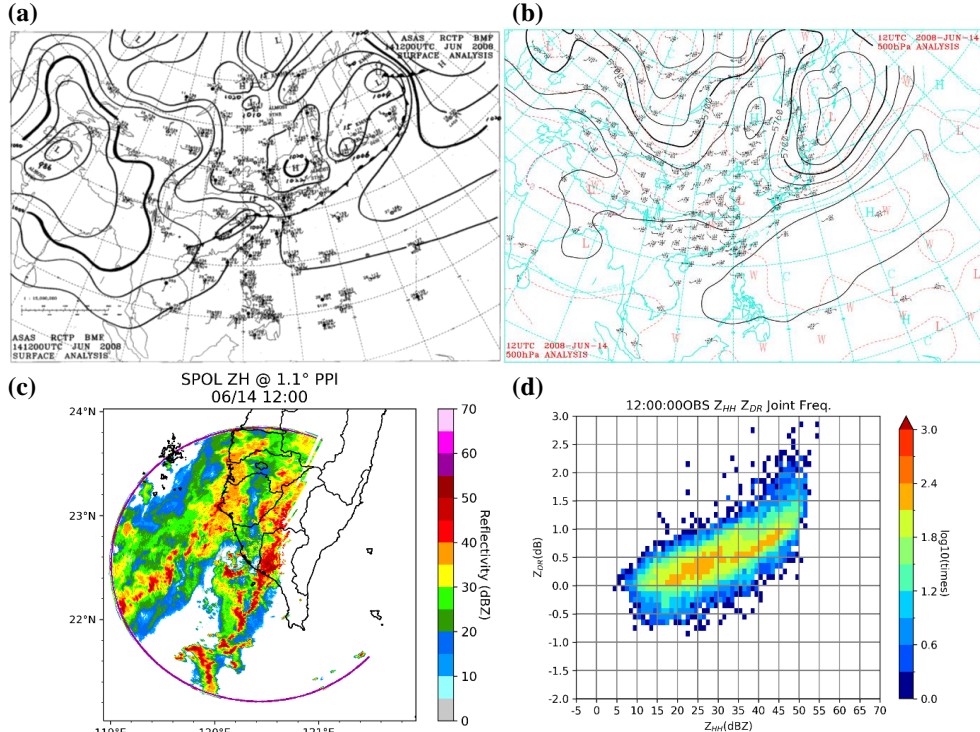

Fig. 1: Synoptic-scale and meso-scale observations of the SoWMEX IOP8 squall line case at 1200 UTC on June 14, 2008. Synoptic-scale: (a) surface weather map and (b) 500 hPa analyses weather map from the Central Weather Administration (CWA). Meso-scale: (c) reflectivity at 1.1˚ elevation, observed by SPOL, and (d) $Z_{HH}$–$Z_{DR}$ joint frequency.



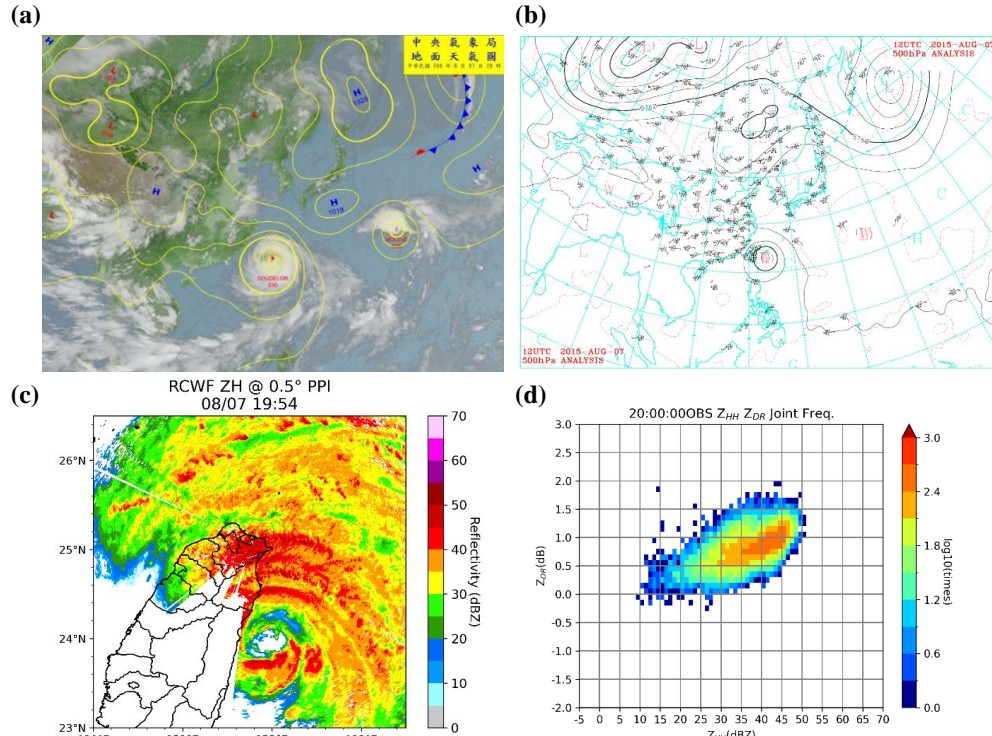

**Fig. 2: Synoptic-scale and meso-scale observations of the Typhoon Soudelor case. Synoptic scale: (a) surface weather map and (b) 500 hPa analyses weather map at 1200 UTC on August 7, 2015, from Central Weather Administration (CWA). Meso scale: (c) reflectivity at 0.5° elevation observed by RCWF, at 1954 UTC and (d) $Z_{HH}$–$Z_{DR}$ joint frequency at 2000 UTC on August 7, 2015.**



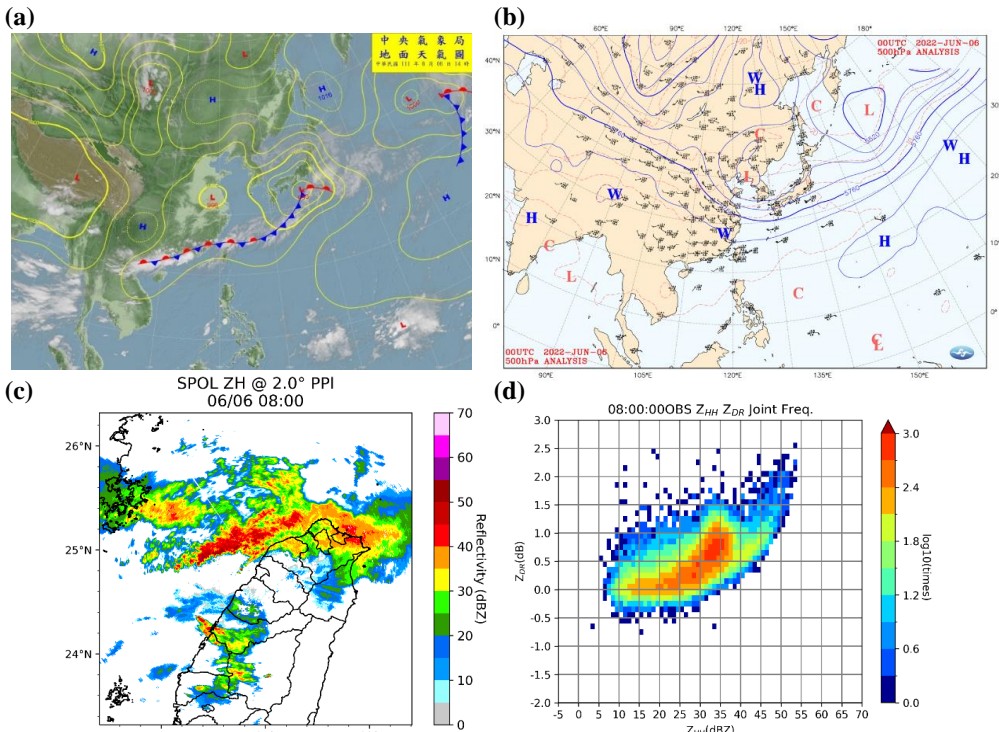

**Fig. 3: Synoptic-scale and meso-scale observations of the TAHOPE IOP3 Mei-Yu frontal case. Synoptic-scale: (a) surface weather map and (b) 500 hPa analyses weather map at 0000 UTC on June 6, 2022, from the Central Weather Administration (CWA). Meso-scale: (c) reflectivity at 2.0° elevation observed by SPOL and (d) $Z_{HH}$ - $Z_{DR}$ joint frequency at 0800 UTC on June 6, 2022.**

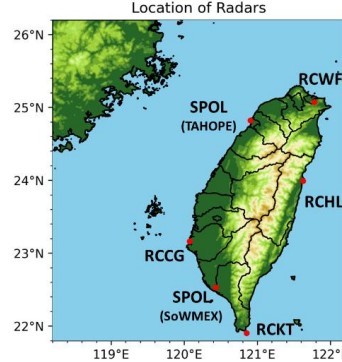


**Fig. 4: Location of the SPOL and operational radars in Taiwan.**



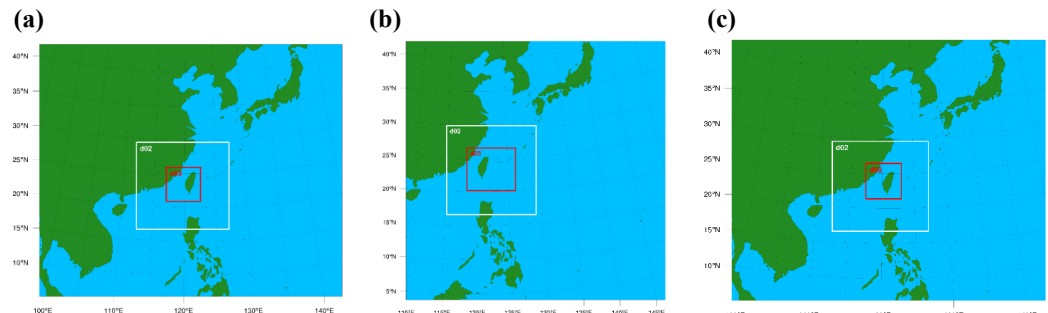

**Fig. 5: Configuration of nested domains in (a) SoWMEX IOP8 squall line case, (b) Typhoon Soudelor case, and (c) TAHOPE IOP3 Mei-Yu frontal case with 15-, 3-, and 1-km horizontal resolution in D01, D02, and D03 respectively.**


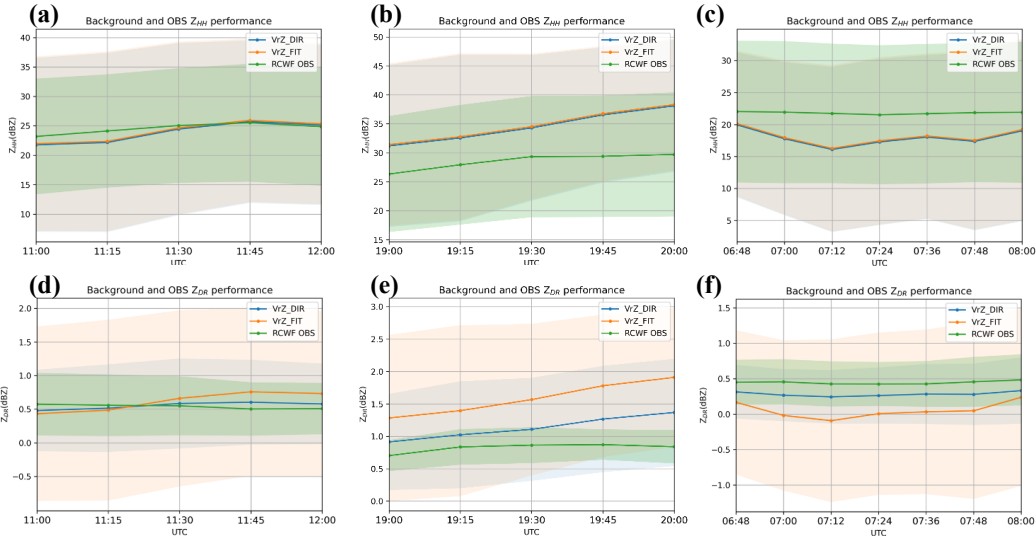

**Fig. 6: Flowchart of the assimilation procedure.**

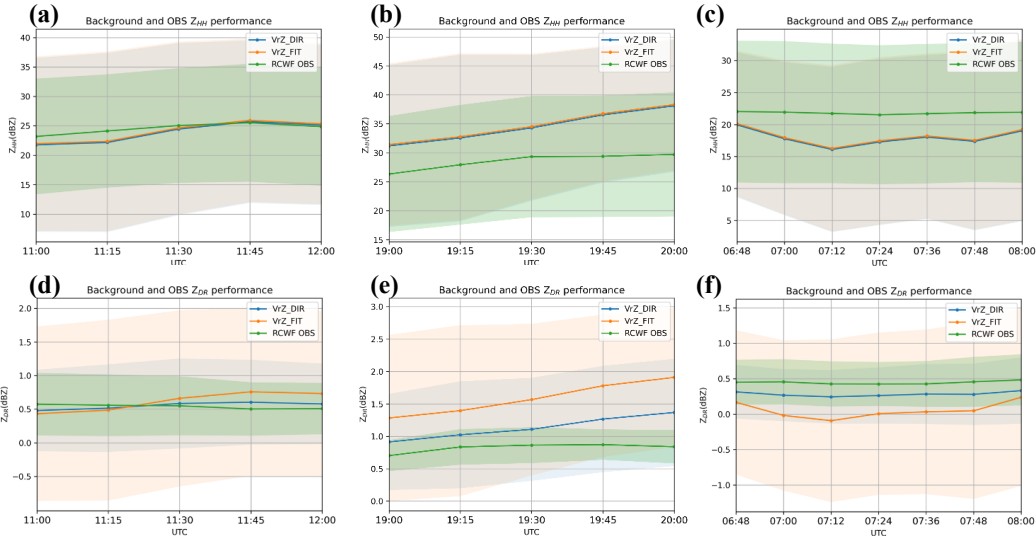

**Fig. 7: Overall performance of (a–c) reflectivity and (d–f) $Z_{DR}$ below 4 km in the VrZ analyses and dual-polarization radar observation in the (a, d) squall line case, (b, e) typhoon case and (c, f) Mei-Yu frontal case. Solid line: mean value of the data; shaded region: within one standard deviation. Blue and orange respectively indicate the VrZ performance when the fitting method and direct integration method are used; green color indicates the observations.**



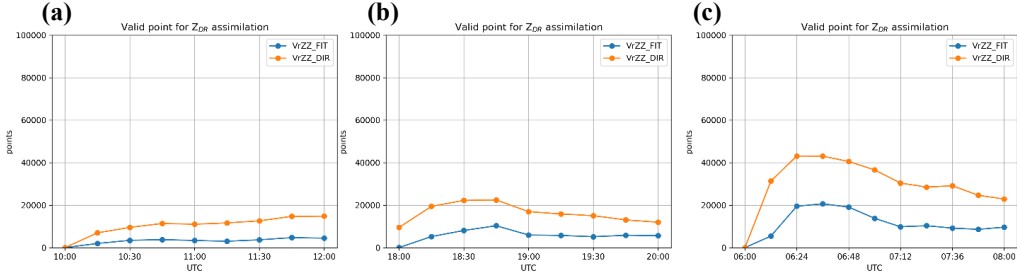

**Fig. 8: Valid points for Z_DR assimilation calculated in VrZ analyses when different operators are used in the (a) squall line case, (b) typhoon case, and (c) Mei-Yu frontal case. Blue and orange respectively indicates the performance when the fitting method and direct integration method are used.**

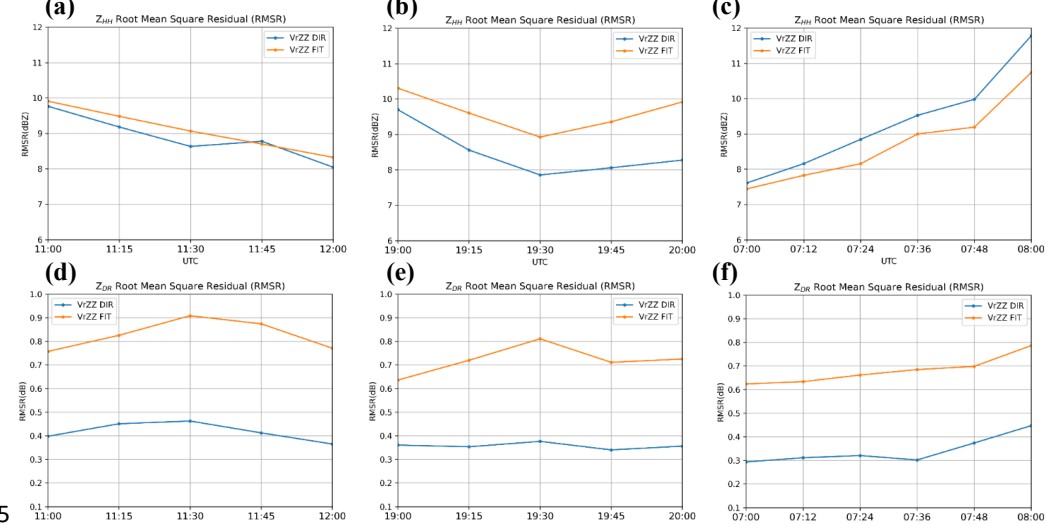

**Fig. 9: RMSR of (a–c) reflectivity and (d–f) Z_DR in the (a, d) squall line case, (b, e) typhoon case, and (c, f) Mei-Yu frontal case.**





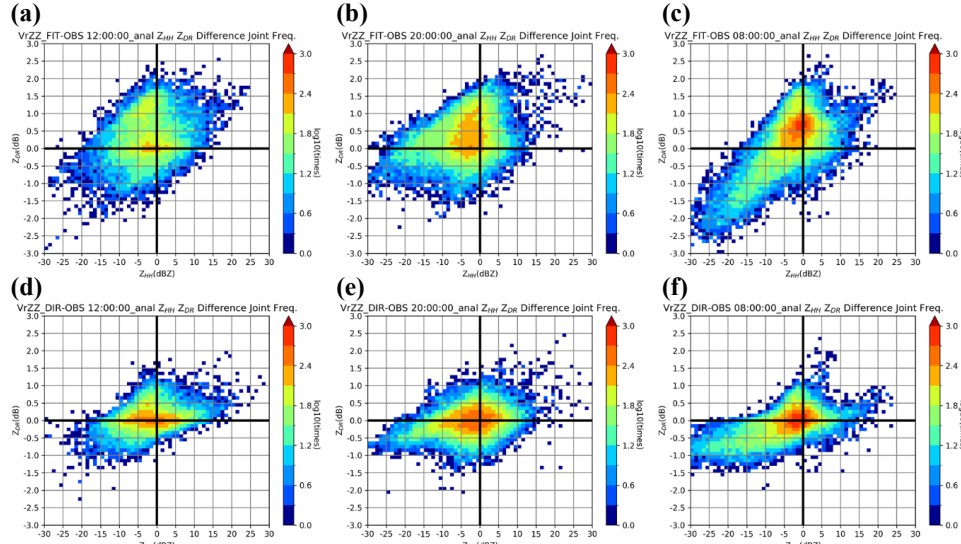

**Fig. 10:** $Z_{HH}$–$Z_{DR}$ **difference joint frequency at the final analyses of the (a–c) VrZZ_FIT and (d–f) VrZZ_DIR**
**experiments for the (a, d) squall line case, (b, e) typhoon case, and (c, f) Mei-Yu frontal case. The shading indicates the**
**occurrence times of the difference calculated from the observation and simulation on the logarithmic scale. The**
**horizontal and vertical axes are the** $Z_{HH}$ **and** $Z_{DR}$ **difference with 1-dBZ and 0.1-dB intervals, respectively.**







**Fig. 11:** *$Z_{HH}$–$Z_{DR}$* **joint frequency at the time of the final analyses in the (a–c) observation, (d–f)VrZZ_FIT background, (g–i) VrZZ_FIT analyses, (j–l) VrZZ_DIR background, and (m–o)VrZZ_DIR analyses for three cases: squall line (left column), typhoon (middle column) and Mei-Yu frontal case (right column). The shading is the occurrence times on the logarithmic scale. The horizontal and vertical axes are the *$Z_{HH}$* and *$Z_{DR}$* with 1-dBZ and 0.1-dB intervals, respectively.**






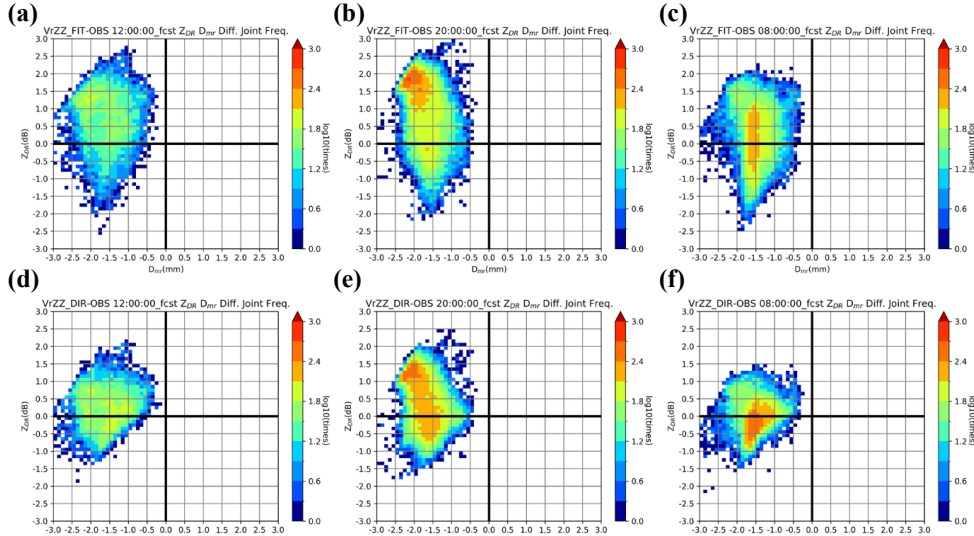

**Fig. 12: Background $D_{mr}$–$Z_{DR}$ joint frequency at the time of the final analyses in the (a–c) VrZZ_FIT and (d–f) VrZZ_DIR experiments for the (a, d) squall line case, (b, e) typhoon case, and (c, f) Mei-Yu frontal case. The shading indicates the occurrence times on the logarithmic scale. The horizontal and vertical axes indicate $D_{mr}$ and $Z_{DR}$ with 0.1-mm and 0.1-dB intervals, respectively.**

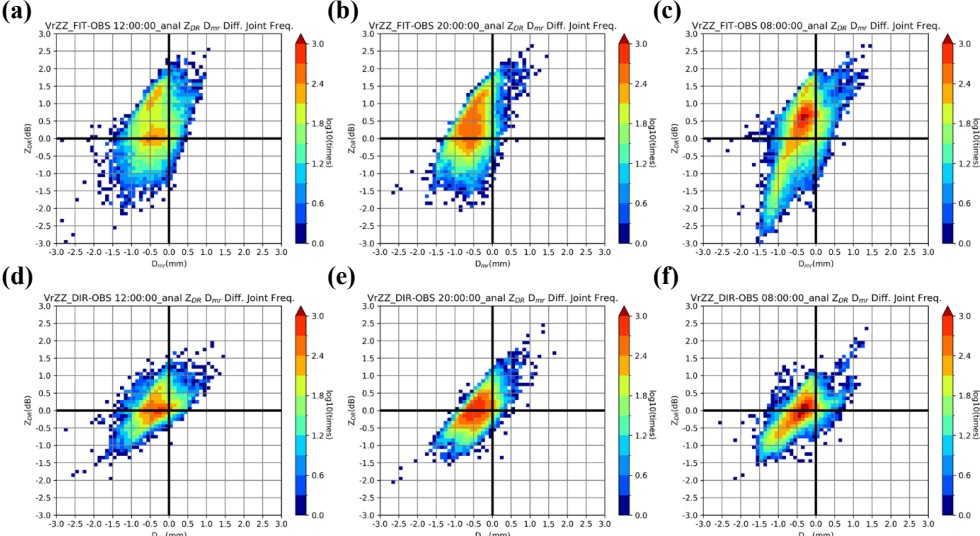

**Fig. 13: Analyzed $D_{mr}$–$Z_{DR}$ joint frequency at the time of the final analyses in (a–c) VrZZ_FIT and (d–f) VrZZ_DIR experiments for the (a, d) squall line case, (b, e) typhoon case, and (c, f) Mei-Yu frontal case. The shading indicates the occurrence times on the logarithmic scale. The horizontal and vertical axes indicate $D_{mr}$ and $Z_{DR}$ with 0.1-mm and 0.1-dB intervals, respectively.**



**Table 1: Configuration of the WLRAS**

| Updated Variables | U, V | $q_v$, T | $q_c$, $q_i$, $N_{Tc}$ | $q_r$, $q_s$, $q_g$, $N_{Tr}$, W |
|---|---|---|---|---|
| Variable Localization | Vr | | Vr $Z_{HH}$ $Z_{DR}$ | |
| Hori. Loc. Radius | 36 km | 24 km | | 12 km |
| Vert. Loc. Radius | 4 km | | | |
| Inflation Factor | 1.08 | | | |


**Table 2: Configuration of the T-matrix scattering simulation**

| T-matrix simulation setting (raindrop) | | | | |
|---|---|---|---|---|
| Range of raindrop size | Bins | Temperature | Canting angle | Axis ratio |
| 0.08 ~ 8 mm | 100 | 10°C | Mean value and standard deviation: 0° | Bradnes et al.(2003) |

**Table 3: Experimental design**

| EXP. | Assi. VARs | Calculation Method |
|---|---|---|
| VrZ | Vr $Z_{HH}$ | Analytical |
| VrZZ_FIT | Vr $Z_{HH}$ $Z_{DR}$ | |
| VrZZ_DIR | | Numerical Integration |