# Peer review of "Evaluate the Impact of Power-Law Scattering Amplitude Fitting on Dual-Polarization Radar Data Assimilation—Summertime Cases Study"

_EGUsphere, 2025_

## Author Comment (AC1)

**General comments:**
This study evaluates the simulation of radar variables ($Z_H$ and $Z_{DR}$) by the polarimetric radar observation operator of Jung et al. (2008) using the power-law fitting and direct integration methods for scattering calculations. This study is interesting for the polarimetric radar data assimilation. However, both the fitting and direct integration methods presented in this manuscript have their own inherent problems, which have already been addressed or improved by Jung et al. (2010), Dawson et al. (2014), Putnam et al. (2019), and Zhang et al. (2021). In short, the fitting method fails to accurately simulate the polarimetric radar signatures, as demonstrated in the manuscript results. Based on the operator of Jung et al. (2008), Jung et al. (2010) developed more accurate and generalized operators using the direct integration method for both rain and ice hydrometeors. Nevertheless, these operators are complex and require computationally expensive numerical integration over the particle size distribution. Subsequently, Putnam et al. (2019) modified the operators of Jung et al. (2010), introducing precomputed lookup tables to increase computational efficiency with some sacrifice to accuracy, and demonstrated their application in assimilating real ZDR (Putnam et al. 2021). However, the operators modified by Putnam et al. (2019) are still computationally expensive and difficult to use in data assimilation, especially in variational assimilation. Zhang et al. (2021) developed a set of parameterized operators based on the numerical integration of the scattering weighted by the particle size distribution. It is challenging to balance the computational efficiency and accuracy of the observation operator within data assimilation systems.
I really appreciate the authors' efforts in exploring a challenging path toward the effective assimilation of polarimetric radar observations. However, I would strongly encourage the authors to find new avenues rather than retreading ground that has already been explored.

We sincerely appreciate the reviewer's considerate comments regarding the inherent issues identified in previous studies, as well as for the valuable suggestions pointing to potential new research directions. To the best of our knowledge, however, the limitations of the power-law fitting approach—particularly the issue of unreasonable negative $Z_{DR}$—have not been explicitly addressed in the existing literature. Despite this deficiency, power-law fitting–based operators continue to be employed in recent studies (e.g., Kabasawa et al., 2018; Lee et al., 2026). Since the bias of the background directly leads to physically unreasonable outcomes after data assimilation, it is essential to highlight such systematic biases and to ensure that their limitations are well recognized by the research community. Although the polynomial fitting method addressed by Zhuang et al. (2021) can alleviate the negative $Z_{DR}$, it still exhibits uncertainties in representing $Z_{DR}$, particularly for wet snow and for extreme value in the real case simulation. Furthermore, since the polynomial fitting is applied after numerical integration, it is needed to first evaluate whether the integration-form operator provides sufficient accuracy under the meteorological conditions in Taiwan before applying the polynomial fitting. While we acknowledge the importance of systematically comparing different operators and discussing their respective strengths and weaknesses, we believe that explicitly identifying significant biases and preventing potential misuse of existing approaches is both necessary and timely. By clarifying these issues, our study aims to provide guidance toward more appropriate choices and to improve the physical consistency of observation operators. Consequently, the background simulations and associated error structures can be made more reliable prior to data assimilation.

**Major comments:**

(1) Introduction: I suggest that the authors systematically review the development of polarimetric radar observation operators, highlighting the respective strengths and weaknesses of different operators in both simulation and assimilation.

Response: We would additionally address the development of observation operators in the revised manuscript. As this study primarily focuses on the differences between power-law fitting approach and the numerical integration approach, we will provide a more in-deep comparison between these two types of operators. The novel method, polynomial fitting method addressed by Mahale et al.(2019) and Zhang et al.(2021), would be briefly depicted in the manuscript as well. The melting model also plays an important role in the observation operators, and it is necessary to modify the melting model toward the observation. Such modification could be found in Dawson et al.(2014) and Zhang et al. (2021). However, because the present study mainly targets the raindrop-dominant region below the melting layer, the remodeling of the melting model would be discussed only briefly. We will try our best to describe the evolution of the operators and point out how important the different steps are.

(2) Methodology: The manuscript appears to employ the direct integration method only for the raindrop. It is unclear how the manuscript handles ice-phase particles (snow and graupel/hail) and mixed-phase particles (wet snow and wet graupel/hail). A fair comparison should use the same method for all hydrometeor species. Additionally, the sacrifice of accuracy in the fitting method is unavoidable. Therefore, the authors need to clarify the computational advantages of this approach. If the fitting method provides neither improved efficiency nor adequate accuracy, then what is the rationale for using it instead of the direct integration method?

Response: We thank the reviewer for highlighting the computational advantages of the fitting-based approach. Based on our experiments, replacing only the raindrop-related term with the numerical integration formulation can reduce the overall computational cost by approximately 80%, resulting in a substantial acceleration of the simulations. However, this modification simultaneously introduces a pronounced negative $Z_{DR}$. In this study, the overall structure of the observation operator follows Jung et al. (2008), while the fitting coefficients are adopted from You et al. (2020), for both liquid-phase and ice-phase terms. Since the purpose of our article is addressing the unavoidable negative $Z_{DR}$ in simulating the raindrop-related terms, we intentionally keep other components of the operator unchanged. This design allows us to isolate the impact of modifying a single term and to clearly highlight the resulting differences. For sure, it is always an important issue that how to well simulate the ice-phase related term inside the operator, and our team would try hard to set up and examine the ice-phase term; these efforts are on the process and we would like to see the results of using the integration form for all hydrometeor species.

(3) Results: The authors need to present the spatial distribution of the polarimetric radar variables simulated by different methods, including both horizontal and vertical cross-sections. In section 5.3, the authors used the fitted Dmr from the radar variables as a reference "observation" for comparison. However, it is unclear what

the purpose of such a comparison is when the "observations" themselves do not represent the truth. Why are the radar variables not compared directly?

Response: The spatial distribution of the dual-polarization radar variables would be presented in the manuscript after the revision. In section 5.2, the simulation of the polarimetric radar variables have been validated by the observed data. However, since dual-polarization radar variables are not prognostic variables in numerical models, during model updating, model state variables can only be adjusted through their correlations with dual-polarization parameters. These correlations are derived from ensemble model simulations, and it remains unknown whether the same relationships can be described by observations. To assess whether these correlations are reliable, microphysical observations can be used—beyond radar measurements—for verification. Yet, the lack of three-dimensional in-situ microphysical observations poses substantial challenges for such validation. Lee et al. (2019) developed fitted relationships using microphysical observations and radar data, aiming to describe long-term statistical connections between radar variables and observed microphysical quantities. Using these retrieval formulas, we can estimate microphysical characteristics from radar observations in regions lacking in-situ measurements. These estimated characteristics can then be used to validate real-case analyses, helping determine whether the microphysical structures—composed of model forecast variables within the analysis fields—become more consistent with observation-derived characteristics after data assimilation within the radar-covered region. In other words, we aim to assess whether the relationships between dual-polarization radar variables and microphysical parameters become more similar to those fitted from observations.

**Reference**

Dawson, D. T., E. R. Mansell, Y. Jung, L. J. Wicker, M. R. Kumjian, and M. Xue: Low-Level ZDR Signatures in Supercell Forward Flanks: The Role of Size Sorting and Melting of Hail, Journal of the Atmospheric Sciences, 71, 276-299, doi: 10.1175/jas-d-13-0118.1, 2014.

Lee, M.-T., P.-L. Lin, W.-Y Chang, B.K Seela and J. Janapati: Microphysical characteristics and types of precipitation for different seasons over north Taiwan, J. Meteorol. Soc. Jpn., 97(4), 841–865, doi: 10.2151/jmsj.2019-048, 2019.

Mahale, V. N., G. Zhang, M. Xue, J. Gao, and H. D. Reeves: Variational retrieval of rain microphysics and related parameters from polarimetric radar data with a parameterized operator. J. Atmos. Ocean. Technol.,36(12), 2483−2500, doi: 10.1175/JTECH-D-18-0212.1, 2019.

Jung, Y., G. Zhang, and M. Xue: Assimilation of Simulated Polarimetric Radar Data for a Convective Storm Using the Ensemble Kalman Filter. Part I: Observation Operators for Reflectivity and Polarimetric Variables, Monthly Weather Review, 136, 2228-2245, doi: 10.1175/2007MWR2083.1, 2008a.

Jung, Y., M. Xue, G. Zhang, and J. M. Straka: Assimilation of Simulated Polarimetric Radar Data for a Convective Storm Using the Ensemble Kalman Filter. Part II: Impact of Polarimetric Data on Storm Analysis, Monthly Weather Review, 136, 2246-2260, doi: 10.1175/2007MWR2288.1, 2008b.

Kawabata, T., T. Schwitall, A. Adachi, H.-S. Bauer, V. Wulfmeyer, N. Nagumo, and H. Yamauchi: Observational operators for dual polarimetric radars in

variational data assimilation systems (PolRad VAR v1.0), Geosci. Model Dev., 11, 2493–2501, doi: 10.5194/gmd-11-2493-2018, 2018.

You, C. R., K. S. Chung, and C. C. Tsai: Evaluating the Performance of a Convection-Permitting Model by Using Dual-Polarimetric Radar Parameters: Case Study of SoWMEX IOP8, Remote Sensing, 12, 3004, doi: 10.3390/rs12183004, 2020.

Zhang, G., J. Gao, and M. Du: Parameterized forward operators for simulation and assimilation of polarimetric radar data with numerical weather predictions. Adv. Atmos. Sci., 38(5), 737−754, doi: 10.1007/s00376-021-0289-6, 2021.

---

## Author Comment (AC2)

**General comment:**
The topic is of great interest and relevance. The direct assimilation of radar data, and in particular polarimetric variables, represents a challenge in data assimilation.

The article refers to various types of radar data assimilation (variational and LEFTK), but does not describe in detail the different methodologies used for the assimilation of such data. For example, with regard to LEFTK, the EMOVRADO operator was developed within the ICON limited area modelling framework, which also has a specific component for "polarimetric data assimilation".

The introductory part of the article should provide an in-depth overview of this, following a logical path from the definition of the observables to be assimilated, the methodologies already in use, the limitations and strengths of the methodologies. In this context, it is unclear what added value the proposed methodology offers over current methods, considering the results obtained.

We sincerely appreciate the reviewer's considerate comments regarding the insufficiency of the Introduction, and we have revised this section to substantially improve its completeness. As the primary objective of this research is to validate and point out the unavoidable biases associated with two types of observation operators, the revised Introduction now focuses more explicitly on studies related to dual-polarization radar observation operators. In addition, key references on the LETKF framework, including comparisons between LETKF and other EnKF-based systems, are now briefly introduced for context. Since the biases of simulated observed variables in the background leads to unreasonable error covariance degrading data assimilation performance, it is needed to validate the behavior of the observation operator itself. The added value of this study is explicitly identifying the unavoidable negative $Z_{DR}$ associated with small raindrops. We believe that by depicting these issues could guide future researchers toward more appropriate operator choices and make the simulation more reasonable.

**Specific comments:**
In some parts, the article is difficult to read as it does not follow a logical sequence in its descriptions and explanations.
For example, Section 2 introduces the LETKF system without ever having explained what it consists of (there is a brief mention of the ETKF). Moreover, it is unclear what it means that the WLTAS system is a deterministic ENKF.

Response: Additional descriptions of the LETKF will be incorporated into the Introduction, including its advantages such as the use of covariance localization to prevent analyses from being contaminated by spurious long-range correlations, the ability to compute analyses independently at each grid point, and the benefits of deterministic EnKF. Compared to the stochastic EnKF perturbing the observation before assimilation, deterministic EnKF don't need to perturb the observation to prevent the sampling error resulting from using the perturbed observation.

From the introduction, it is unclear whether this study is conducted for all types of hydrometeors or whether, as it appears to be, it is conducted only for raindrops. If it is a choice, the reason must be given.

Response: In this study, we employ the same configuration for the ice-particle-related terms in the observation operator, using the power-law fitting approach based on Jung et al. (2008a) in all experiments. For the raindrop-related terms, however, we apply two different formulations—power-law fitting and direct numerical integration—to compute the polarimetric radar variables. The reason for modifying only the raindrop-related terms is to keep the control factors as simple as possible. Because the primary objective of this study is to investigate the unavoidable negative $Z_{DR}$ introduced by the power-law fitting approach for raindrop-related terms, isolating the impact of this single modification allows the errors to be identified more directly and unambiguously. For sure, it is always an important issue that how to properly simulate the ice-phase related term inside the operator, and our team would try hard to set up and examine the ice-phase term. At the same time, it should be noted that validation of ice-particle simulations is particularly challenging in Taiwan due to the limited availability of in-situ airborne observations. For these reasons, the present study focuses on raindrop-related terms and highlights the associated biases below the melting layer, which we consider an essential first step before extending the analysis to ice-phase processes in future work.

In section 3, it would be preferable to first describe the types of data used and then the selected case studies

Response: We agree that, in general, it is preferable to introduce the observational data prior to presenting the case descriptions. However, in this study, the scanning strategies of the operational radars—including elevation configurations and single- versus dual-polarization modes—differ among the three cases, as the radar systems have been gradually upgraded over the past decades. To clearly describe the scanning strategies relevant to each case, we therefore first introduce the cases themselves, followed by a detailed description of the corresponding radar observations.

In section 4, in the part concerning model configuration, it is assumed that the reader is already familiar with the specifications of the model used.

Response: To ensure that readers can clearly understand the background of our model configuration, a brief description of the WRF model and the microphysics parameterization scheme has been added to the manuscript. In addition, the model domain coverage is now described to provide clearer context for the simulations. We have made every effort to present the model configuration in a concise and accessible manner for the readers.